# Lysophosphatidic Acid Receptors LPAR5 and LPAR2 Inversely Control Hydroxychloroquine-Evoked Itch and Scratching in Mice

**DOI:** 10.3390/ijms25158177

**Published:** 2024-07-26

**Authors:** Caroline Fischer, Yannick Schreiber, Robert Nitsch, Johannes Vogt, Dominique Thomas, Gerd Geisslinger, Irmgard Tegeder

**Affiliations:** 1Institute for Clinical Pharmacology, Faculty of Medicine, Goethe-University Frankfurt, 60590 Frankfurt am Main, Germany; line_bittner@yahoo.de (C.F.); thomas@med.uni-frankfurt.de (D.T.); geisslinger@em.uni-frankfurt.de (G.G.); 2Fraunhofer Institute for Translational Medicine and Pharmacology ITMP, 60596 Frankfurt am Main, Germany; yannick.schreiber@itmp.fraunhofer.de; 3Institute for Translational Neuroscience, Medical Faculty, WWU Münster, 48149 Münster, Germany; nitschr@uni-muenster.de; 4Department of Molecular and Translational Neurosciences, Institute for Anatomy and Center of Molecular Medicine Cologne (CMMC), and Cologne Excellence Cluster for Aging associated Diseases (CECAD), University of Cologne, 50923 Köln, Germany; johannes.vogt@uk-koeln.de; 5Fraunhofer Cluster of Excellence of Immune Mediated Diseases (CIMD), 60596 Frankfurt am Main, Germany

**Keywords:** lysophosphatidic acid, pruritus, itch, plasticity related gene 1, autotaxin, dorsal root ganglia, hydroxychloroquine, skin

## Abstract

Lysophosphatidic acids (LPAs) evoke nociception and itch in mice and humans. In this study, we assessed the signaling paths. Hydroxychloroquine was injected intradermally to evoke itch in mice, which evoked an increase of LPAs in the skin and in the thalamus, suggesting that peripheral and central LPA receptors (LPARs) were involved in HCQ-evoked pruriception. To unravel the signaling paths, we assessed the localization of candidate genes and itching behavior in knockout models addressing LPAR5, LPAR2, autotaxin/ENPP2 and the lysophospholipid phosphatases, as well as the plasticity-related genes Prg1/LPPR4 and Prg2/LPPR3. LacZ reporter studies and RNAscope revealed LPAR5 in neurons of the dorsal root ganglia (DRGs) and in skin keratinocytes, LPAR2 in cortical and thalamic neurons, and Prg1 in neuronal structures of the dorsal horn, thalamus and SSC. HCQ-evoked scratching behavior was reduced in sensory neuron-specific Advillin-LPAR5^−/−^ mice (peripheral) but increased in LPAR2^−/−^ and Prg1^−/−^ mice (central), and it was not affected by deficiency of glial autotaxin (GFAP-ENPP2^−/−^) or Prg2 (PRG2^−/−^). Heat and mechanical nociception were not affected by any of the genotypes. The behavior suggested that HCQ-mediated itch involves the activation of peripheral LPAR5, which was supported by reduced itch upon treatment with an LPAR5 antagonist and autotaxin inhibitor. Further, HCQ-evoked calcium fluxes were reduced in primary sensory neurons of Advillin-LPAR5^−/−^ mice. The results suggest that LPA-mediated itch is primarily mediated via peripheral LPAR5, suggesting that a topical LPAR5 blocker might suppress “non-histaminergic” itch.

## 1. Introduction

Lysophosphatidic acids are versatile lipid-signaling molecules. The most abundant candidates are LPA16:0, 18:1, 18:2 and 20:4, with likely distinct functions depending on the chain length and saturation, as well as on receptor affinities [1]. The research on LPA functions has focused on extracellular LPAs, which are generated from phosphatidic acid by phospholipases A [2,3] or from lysophospatidylcholine (LPC) via the phospholipase D, autotaxin (ectonucleotide pyrophosphatase/phosphodiesterase 2 (ENPP2) [4,5]. LPAs activate G protein-coupled LPA receptors, LPAR1-5, and the atypical LPAR6, which is similar to purine receptors [6]. Degradation and transport involves lysophospholipid phosphatases, LPPs [7]. The subtypes 4 and 3 (LPPR4, LPPR3), also known as plasticity-related genes 1 and 2 (Prg1, Prg2), control LPA levels in the CNS at postsynaptic sites (Prg1) via internalization, thereby regulating synaptic strength, or in axonal compartments (Prg2), protecting axons from repelling cues [8,9,10]. Peripheral LPAs are involved in signaling process of cell mobility [11] and migration [12], angiogenesis [13], platelet aggregation [14,15,16], inflammation [17,18,19,20], intestinal permeability [21] and wound healing [22,23]. Complexity arises from cell type and tissue specific expression of distinct LPAR subtypes [24] and signaling via Gi, Gs or Gq. Recent reviews have addressed this topic [20,25,26,27]. The current view of the complex LPA/LPAR system is mostly based on non-site-specific conventional knockout mice [28,29,30,31] and more-or-less specific receptor-subtype agonists and antagonists, which often have unknown in vivo pharmacokinetic properties [32,33].

In the nervous system, LPAR1 is mainly expressed by myelinating glia [34,35,36], LPAR2 in presynaptic glutamatergic terminals [8,37], LPAR3 possibly in astrocytes [38], LPAR4 at the blood–brain barrier and LPAR5 in peripheral sensory neurons [39], suggesting that LPAR5 is involved in somatosensory functions [40,41]. Indeed, results obtained with general LPAR5 knockout mice suggested that LPAR5 mediates LPA-dependent neuropathic nociception and pruriception [39,42]. Ex vivo studies revealed that the serum of patients with cholestatic itch evoked calcium responses in neuronal cell lines [43]. LPAs were elevated in these serum samples, and injection of LPA into the mouse skin evoked a scratching response, suggesting that LPAs caused cholestatic itch. Further studies showed that LPAs cause a release of histamine from ex vivo skin fragments [44] and contribute to the development of mast cells [45,46]. However, subsequent studies did not unequivocally show that LPAs facilitate or enhance histamine- or neuropeptide-evoked itch [47].

A study in humans revealed that LPA microinjections into the skin elicited burning pain, whereas LPA application via cowhage spicules evoked moderate itch [48], suggesting a differential LPA-mediated activation of nociceptive and pruriceptive nerve fibers. The sensation of itch is mediated by slowly conducting sensory neurons with unmyelinated C-nerve fibers that are also crucial for nociception [49]. Experimental studies in mice suggest that distinct neurons are specialized pruriceptors [50] and are characterized by the expression of itch-sensing receptors of the mas-related G protein receptor (Mrgpr) family. These neurons are particularly responsive to hydroxychloroquine or the neuropeptide BAM 8–22 [51,52], both of which are considered non-histaminergic pruritogens. Histamine-evoked itch is elicited experimentally via the compound 48/80, which causes mast cell degranulation [53,54] and secondary release of prurigenic mediators, including proteases, histamine and serotonin. Therefore, Cp 48/80 is regarded as a histamine-dependent pruritogen. Owing to antibody limitations, it is currently not known which DRG neuron subtypes express which types of LPAR.

To further assess the functions and putative therapeutic implications of LPAs for itch, we studied if, where and when they are released and found an increase in LPAs in the skin and thalamus, i.e., at the site of sensation, and crucial processing of itch in the somatosensory axis. The result suggested that LPAs are peripheral and central mediators of itch signaling. To dissect out the functional implications and receptors, we studied the expression, localization and itch response upon stimulation with HCQ or Cp 48/80 in knockout models addressing LPAR5, LPAR2, Prg1, Prg2 and autotaxin. The selection was based on previous work [55,56,57,58,59,60,61,62] and mRNA studies showing expression of these candidates at crucial sites (Figure 1). The results revealed the pro-pruriceptive effects of peripheral LPAR5 but suggested itch inhibitory LPA-signaling via LPAR2.

## 2. Results

### 2.1. Itch Stimuli Increase LPAs in Skin and Thalamus

In the first set of experiments, we assessed where and when LPAs are released along the itch-signaling axis upon stimulation (Figure 1A–C) and the localization of LPA receptors (LPAR2, LPAR5; Figure 2, Figure 3 and Figure 4) and transporters (Prg1, Figure 5) at crucial synaptic sites.

Itch stimulation increased LPAs in the thalamus (Figure 1A) after both stimuli and in the skin after HCQ (Cp 48/80 not assessed) (Figure 1C), but not in plasma (Figure 1B). The LPA increase in the thalamus was stronger upon stimulation with hydroxychloroquine (HCQ) than Cp 48/80, suggesting that LPAs were involved primarily in non-histaminergic itch. Therefore, HCQ was used to assess LPAs in the skin. The release pattern suggested that LPAs in the skin would lead to activation of itch-sensitive DRG neurons, and LPAs in the thalamus affect the thalamic control of cortical stimulation. We assessed the expression and localization of “itch-relevant” candidate genes via QRT-PCR at crucial sites (Figure 1D). The expression patterns show a high LPAR5 abundance in the DRGs and skin but not in the CNS, whereas LPAR2, which has been recognized as synaptic enhancer in other systems, was abundantly expressed in the CNS. Prg1 and Prg2 are CNS-specific lipid phosphate phosphatases. Prg2 showed high expression in the spinal cord. For comparison, the mas-related G protein-coupled receptor subtypes were expressed in the periphery and the CNS. Autotaxin, which generates extracellular LPA, had high expression in the spinal cord (SC), and alkylglycerol monooxygenase (AGMO), which contributes to the pool of intracellular LPAs [63], was evenly expressed at a low level.

### 2.2. Localization of LPAR5, LPAR2 and Prg1 along the Itch-Signaling Axis

The experiments above suggested that LPAR5 is a relevant LPA receptor of somatosensory neurons, and that LPAR2 is localized at crucial synaptic sites, in agreement with previous studies. We used LacZ reporter mice to show the expression patterns of LPAR5 (Figure 2 and Figure 3). X-Gal histology and immunofluorescence studies of beta galactosidase showed strong LPAR5 expression in keratinocytes (Figure 2A,B). We did not see LPAR5 on the terminal nerve fibers in the skin, but a strong X-Gal signal revealed the expression of LPAR5 in individual neurons in the DRGs (Figure 3A). Co-immunostaining studies with markers of sensory neurons suggest that LPAR5 was mainly expressed in P2X3- and/or TRPV1-positive DRG neurons, which are small neurons with unmyelinated C-fibers. There was no or low co-expression with the substance P (SP) or the calcitonin gene-related peptide (CGRP) (peptidergic).

**Figure 2 ijms-25-08177-f002:**
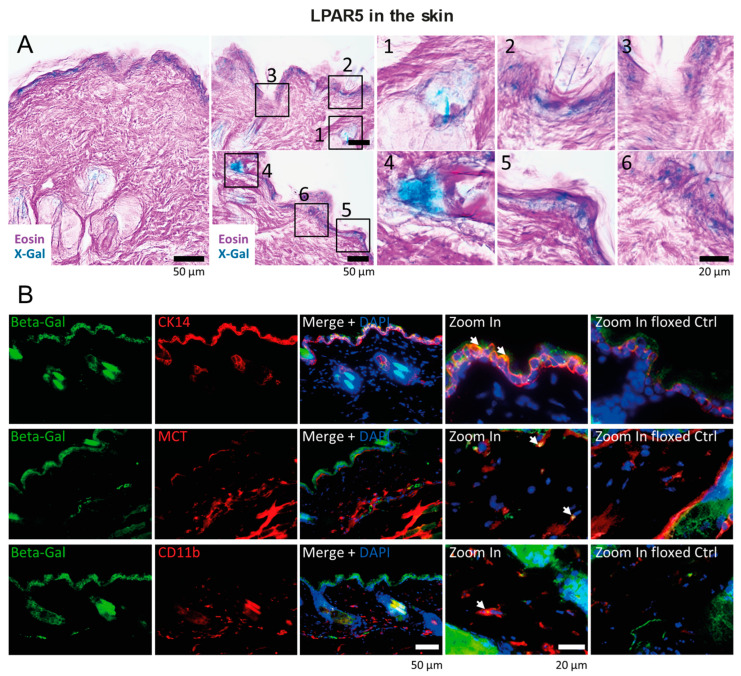
LPAR5 reporter gene expression and localization in the skin. (**A**) X-Gal (blue) histology of skin sections in naïve LPAR5-β-Galactosidase reporter mice. The numbered rectangles show the regions used for zoom-in images. (**B**) β-Galactosidase immunofluorescence in the skin of naïve LPAR5-β-Galactosidase reporter mice. LPAR5-flfl mice were used as negative control. Cytokeratin 14 (CK14) was used as marker for keratinocytes, mast cell tryptase (MCT) for mast cells and cluster of differentiation CD11b for macrophages. DAPI is a nuclear counterstain. Eight LPAR5-LacZ mice were used to assess LPAR5 localization in the skin. White arrows point to double labeled cells.

**Figure 3 ijms-25-08177-f003:**
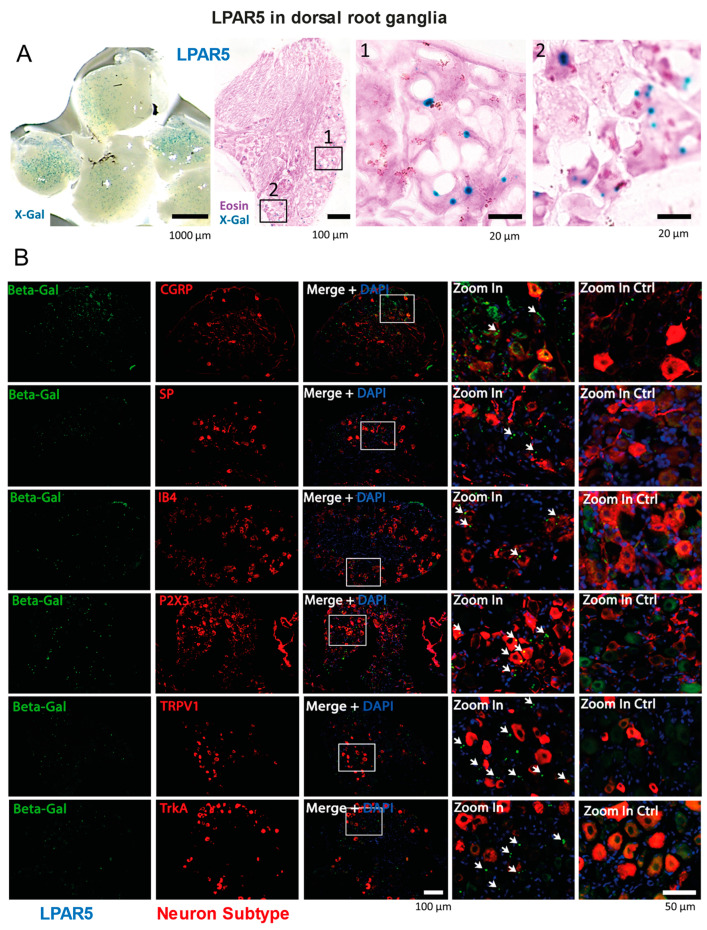
LPAR5 reporter gene expression and localization in DRGs. (**A**) X-Gal histology of the DRGs in naïve LPAR5-β-Galactosidase reporter mice. Ten LPAR5-LacZ mice were used to assess LPAR5 localization in the DRGs and other tissues. Numbered rectangles show the regions used for zoom-in images. (**B**) β-Galactosidase immunofluorescence in the DRGs of naïve LPAR5-β-Galactosidase reporter mice. LPAR5-flfl mice were used as negative control. Neuron subtypes showing LPAR5 reporter expression were assessed by co-immunofluorescence studies with markers for DRG subpopulations. CGRP, calcitonin gene-related peptide (peptidergic); SP, substance P (peptidergic); IB4, isolectin B4 (C-fiber glutamatergic); P2X3, purine receptor (C-fiber, purinergic); TRPV1, transient receptor potential channel (heat sensitive, nonmyelinated); TrkA, tyrosine kinase A (NGF-responsive). The rectangles show the area used for zoom-in and the arrow heads in the zoom-in images point to double positive immunoreactive spots.

The LPAR2 expression in the spinal cord and brain was assessed by RNAscope. The reporter mice are not available. We compared the RNA signal of LPAR2 with positive and negative assay controls (Figure 4). The images revealed abundant LPAR2 expression in the cortex, paraventricular zone and dentate gyrus of the hippocampus, as well as in a few neurons in the dorsal horn of the spinal cord.

**Figure 4 ijms-25-08177-f004:**
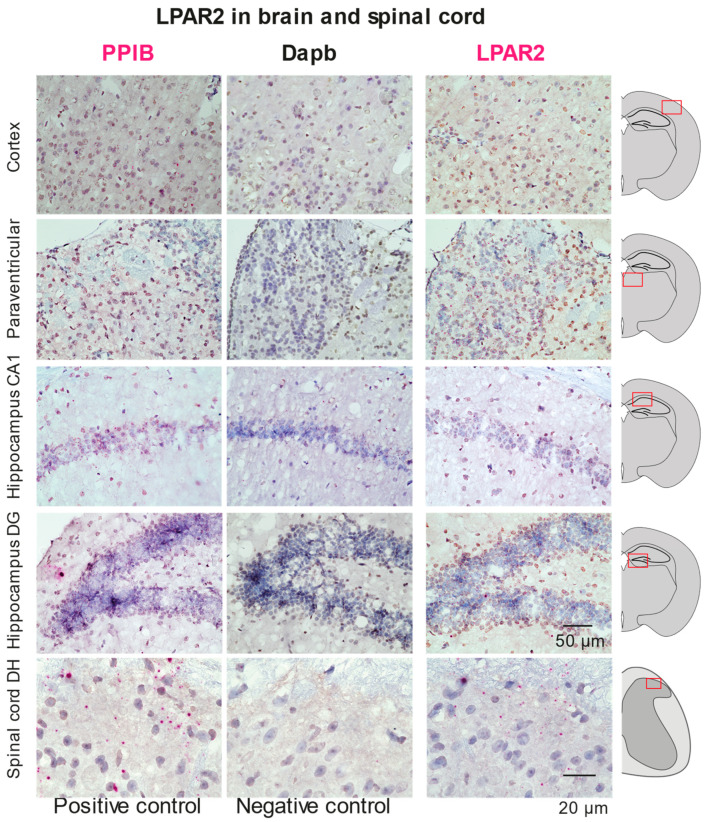
LPAR2 mRNA expression in brain regions by RNA scope technology. For each site, two exemplary images of two mice are shown for LPAR2 in comparison with the housekeeping positive control gene PPIA (Peptidylprolyl Isomerase A) and the negative control bacterial gene, dapB (dipicolinate reductase). Slides were developed with a colorimetric red HRP substrate, and positive reactions occur as red dots. Slides were counterstained with hematoxylin (blue). RNAscope studies were performed on three wildtype mice. Red rectangles indicate the region from where the images were taken.

For Prg1 (LPPR4), we used heterozygous LacZ reporter mice (Figure 5). X-Gal and beta galactosidase immunofluorescence studies revealed strong expression in the somatosensory cortex (Figure 5A), thalamus (Figure 5B) and in Lamina I and II of the dorsal horn (Figure 5C,D), but no expression in the DRGs (Figure 5E). The results are in agreement with previous studies [8]. Immunofluorescent co-expression studies in the dorsal horn (Figure 5C,D) show a dot-like localization of beta galactosidase immunofluorescence in neurons but not in IBA1 + glia. Specific antibodies for Prg2 are currently not commercially available; however, its expression was described to mainly occur at embryonic stages [10,64].

**Figure 5 ijms-25-08177-f005:**
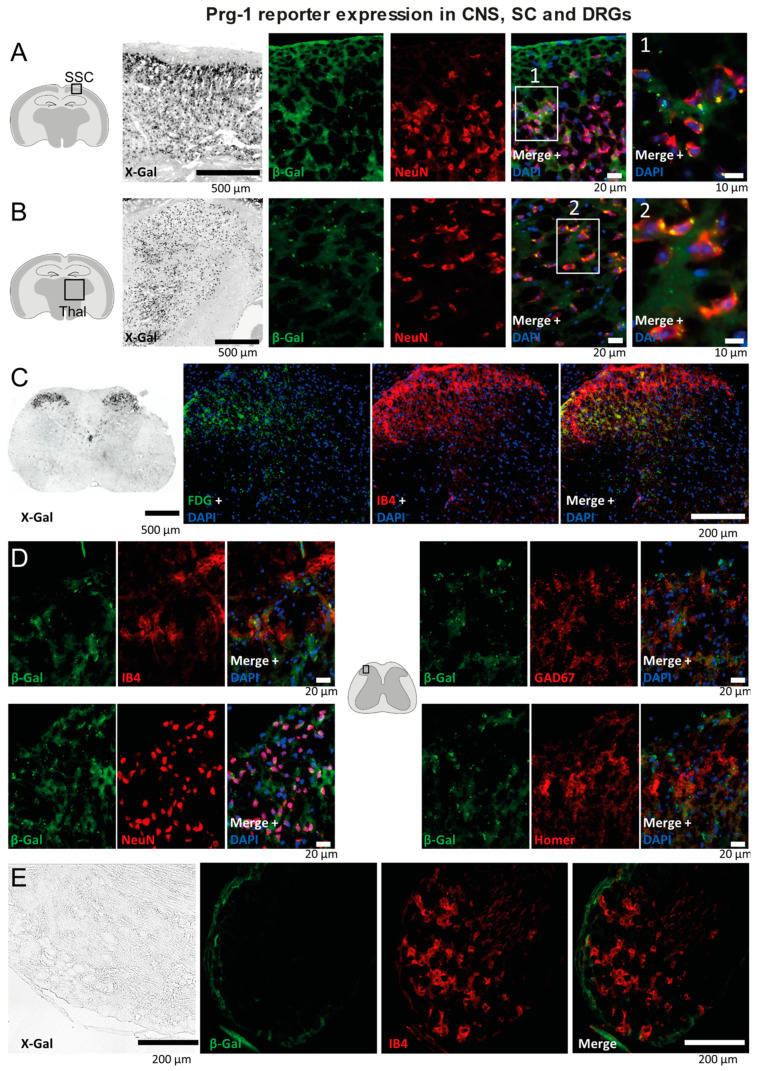
Prg1/LPPR4 reporter gene expression in the CNS, spinal cord and DRGs. (**A**,**B**) X-Gal histology and β-Galactosidase immunofluorescence of the somatosensory cortex (**A**) and thalamus (**B**) in naïve Prg1-β-Galactosidase reporter mice. Neurons were counterstained with the nuclear neuronal marker NeuN (SSC, thalamus). (**C**,**D**) X-Gal histology and β-Galactosidase immunofluorescence in the spinal cord dorsal horn in naïve Prg1-β-Galactosidase reporter mice. Subpanel C shows overviews of the dorsal horn. FDG (Fluorescein di-β-D-galactopyranoside) was used for detection of β-Gal. Isolectin B4 (IB4) was used to highlight Lamina II nerve fiber terminals. Subpanel D shows higher magnifications of the dorsal horn with counterstaining of IB4, NeuN (neurons), Gad67 (GABAergic synapses) and Homer (glutamatergic synapses). Prg1-β-Gal dots reveal synaptic localization. (**E**) X-Gal histology and β-Galactosidase immunofluorescence in the DRGs. No specific signal was observed. DRGs were negative. Eleven Prg1-LacZ mice were used to assess the localization of Prg1 along the itch-signaling axis.

### 2.3. Itching Behavior in Knockout Models

The LPA analyses upon itch stimulation suggested that HCQ leads to LPA release in the vicinity of primary nerve terminals in the skin and at central itch-signaling sites. Considering the histologic receptor localizations, we hypothesized that LPA in the periphery activates LPAR5, and that LPA in the thalamus and cortex activates LPAR2. We further hypothesized that both contributed to pruriceptive sensation and response, which would be modulated by the Prg1-mediated synaptic reuptake of LPA. To address this hypothesis experimentally, we studied itch behavior in knockout (Figure 6 and Figure 7) and pharmacologic models (Figure 8). We focused on the LPAR2 and LPAR5 subtypes of LPA receptors because LPRA1, LPAR3 and LPAR4 are mainly expressed by non-neuronal cells.

### 2.4. Reduced Itch in LPAR5 Knockout but Increased Itch in LPAR2 Knockout Mice

Scratching behavior was elicited by intradermal injection of Cp 48/80, HCQ and BAM 8–22 to observe histaminergic, non-histaminergic and peptidergic itch. The LPAR2^−/−^ mice showed stronger scratching behavior than the LPAR2^+/+^ mice upon injection of HCQ and BAM 8–22 (Figure 6). There was no difference between genotypes upon stimulation with Cp 48/80 (Figure 6).

For LPAR5, we used mice carrying a specific deletion of LPAR5 in sensory neurons. These Advillin-LPAR5^−/−^ mice showed a significantly reduced HCQ-evoked scratching response (Figure 6). The Cp 48/80 responses did not differ significantly from the LPAR5-flfl control mice (Figure 6).

The results show excitatory LPAR5-mediated itch signaling, but inhibitory effects of LPAR2 in the context of itch, which appears to be different from its effects in the hippocampus. Previous electrophysiology studies in the hippocampal in LPAR2^−/−^ slices had revealed that excitatory postsynaptic currents were rather reduced depending on the age of mice at the time of slice preparation [57].

**Figure 6 ijms-25-08177-f006:**
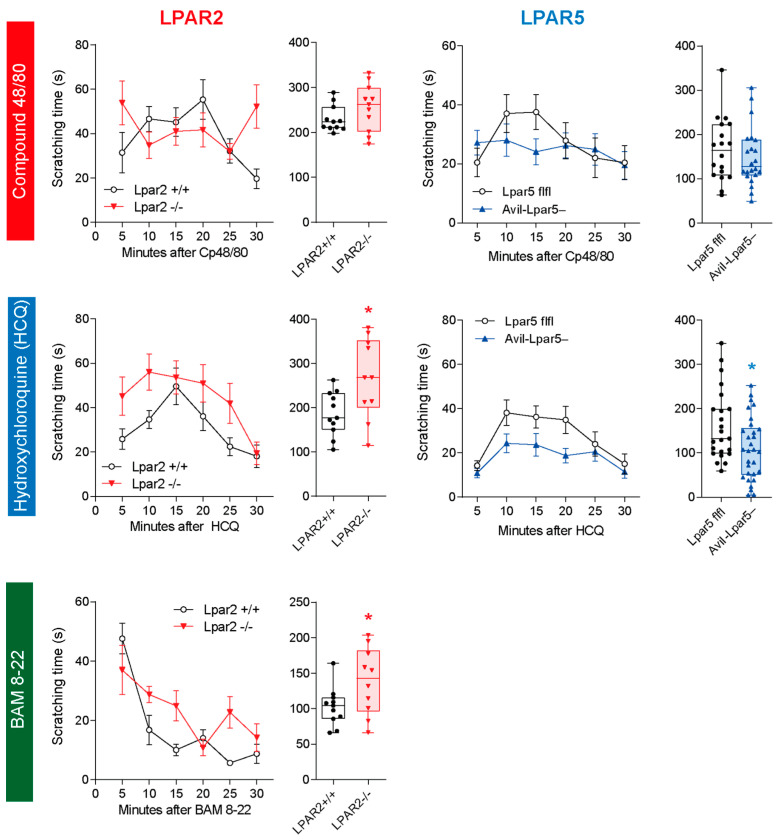
Itch behavior in LPAR2^−/−^ and Advillin-LPAR5^−/−^ mice. Scratching behavior was evoked by intradermal injection of Cp 48/80 (histamine-evoked itch), HCQ (non-histaminergic itch) and Bam 8–22 (peptidergic itch) and observed for 30 min in 5 min intervals. Column headlines indicate the mouse line, and row headlines show the stimulus. Time courses are shown in the left panels (mean ± sem), and box plots show the total scratching time. The box is the interquartile range, the line the median, whiskers minimum to maximum. Scatters show individual mice (n = 11 for LPAR2 experiments; n = 18–22 for LPAR5 experiments). Total scratching times were compared by unpaired, 2-tailed Student’s *t*-test; * *p* < 0.05.

### 2.5. Increased Itch in Prg1 Knockout Mice but Mostly No Effect of Prg2 Knockout

To further address the observed duality of the LPA receptor-dependent itch signaling, we studied knockouts of the postsynaptic LPA-interacting molecules, Prg1 and Prg2 (Figure 7).

Postsynaptic Prg1 is localized at synapses with pre-synaptic LPAR2-mediated enhancement of glutamatergic excitatory currents, according to previous studies [8]. In line with the itching behavior of LPAR2^−/−^ mice, we observed stronger scratching responses in Prg1^−/−^ mice as compared to Prg1^+/+^ control mice upon stimulation with HCQ (Figure 7). Again, the responses to Cp 48/80 were similar in both genotypes and BAM 8–22 differences did not reach statistical significance (Figure 7). The behavior of Prg2^−/−^ mice was equal to the behavior of Prg2^+/+^ mice except for the HCQ response, which was stronger in the knockouts (Figure 7).

**Figure 7 ijms-25-08177-f007:**
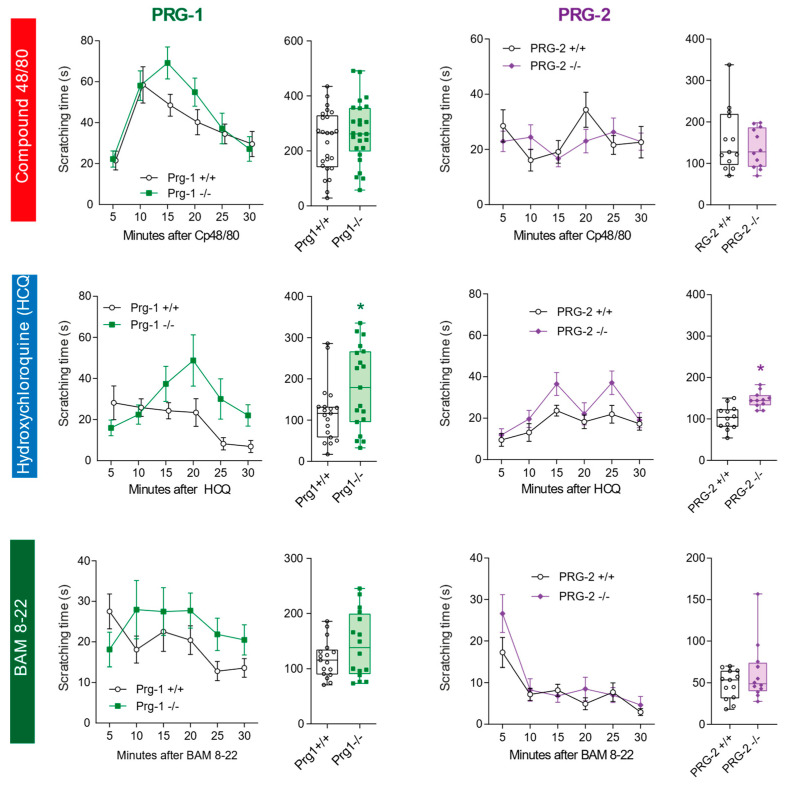
Itch behavior in Prg1/LPPR4^−/−^ and Prg2/LPPR3^−/−^ mice. Scratching behavior was evoked by intradermal injection of Cp 48/80 (histamine-evoked itch), HCQ (non-histaminergic itch) and Bam 8–22 (peptidergic itch) and observed for 30 min at 5 min intervals. Column headlines indicate the mouse line, and row headlines show the stimulus. Time courses are shown in the left panels (mean ± sem), and box plots show the total scratching time. The box is the interquartile range, the line the median, whiskers minimum to maximum. Scatters show individual mice (n = 16–26 for Prg1 experiments; n = 12 for Prg2 experiments). Total scratching times were compared by unpaired, 2-tailed Student’s *t*-test; * *p* < 0.05.

### 2.6. Reduced Itch with Pharmacologic Inhibition of Autotaxin and or LPAR5

The behavioral results in the knockout models suggested a pro-pruriceptive effect of peripheral LPA via LPAR5 but rather an itch-inhibitory effect of central LPAR2/Prg1 synapses. To further address the effects of peripheral versus synaptic LPA, we generated mice carrying a deletion of autotaxin (ENPP2) in astrocytes, driven by GFAP. Astrocytes are a major source of synaptic LPA. These GFAP-Enpp2^−/−^ mice showed similar scratching responses to those of the Enpp2-flfl control mice both upon injection of Cp 48/80 and of HCQ (Figure 8A). In contrast, inhibition of the enzymatic activity of autotaxin with the drug PF8380 strongly reduced both Cp 48/80- and HCQ-evoked scratching behavior (Figure 8B), again showing that peripheral LPA species enhance itch.

**Figure 8 ijms-25-08177-f008:**
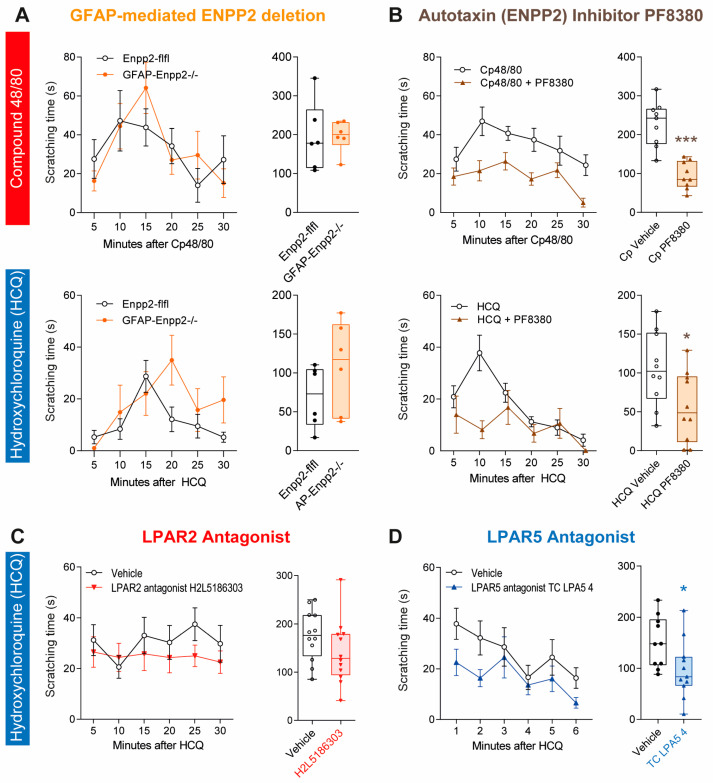
Itch behavior in GFAP-ENPP2^−/−^ mice and effects of autotaxin, LPAR2 and LPAR5 inhibitors. Scratching behavior was evoked by intradermal injection of Cp 48/80 (histamine-evoked itch) or HCQ (non-histaminergic itch) and observed for 30 min at 5 min intervals. Time courses are shown in the left panels (mean ± sem), and box plots show the total scratching time. The box is the interquartile range, the line the median, whiskers minimum to maximum. Scatters show individual mice. Total scratching times were compared by unpaired, 2-tailed Student’s *t*-test; * *p* < 0.05. Drugs were injected 20 min before itch stimulation. Control mice received an equal volume of vehicle (5% DMSO in saline). (**A**) Scratching behavior of GFAP-ENPP2^−/−^ versus floxed control mice (ENPP2-flfl), n = 6 per group. GFAP-ENPP2^−/−^ mice carry a deletion of autotaxin/ENPP2 specifically in astrocytes. (**B**) Scratching behavior in mice treated with the autotaxin inhibitors, PF-8380 or vehicle, i.p., n = 10 per group. (**C**) Scratching behavior in mice treated with the LPAR2 antagonist H2L5186303 or vehicle, i.p., n = 10 or 12 per group. (**D**) Scratching behavior in mice treated with the LPAR5 antagonist TC LPA5 4 or vehicle, i.p., n = 10 per group. * *p* < 0.05, *** *p* < 0.001.

The final set of itch studies assessed if pharmacologic inhibition of LPAR5 mimicked the anti-itch effects of LPAR5 deletion, and if LPAR2 inhibition mimicked the pro-itch effects of LPAR2 deletion. The latter was not the case. Indeed, there was no difference between the vehicle- and H2L5186303 (LPAR2 antagonist)-treated mice (Figure 8C). There was also no effect of the LPAR2 agonist GRI 977143 (Appendix A), but the LPAR5 antagonist reduced the early HCQ-evoked itch response and significantly reduced the overall scratching time, as compared to the vehicle-treated mice.

Since the LPAR2 targeting drugs had no effect, we assessed the plasma and tissue concentrations of the LPAR2 agonist (Appendix A) and the LPAR2 antagonist (Appendix A) to address the pharmacokinetic aspects. The drugs were bioavailable upon oral administration and reached maximum plasma concentrations at/around the time of maximum itch. the dose-adjusted concentrations of agonist and antagonist were similar. However, the brain concentrations of the LPAR2 antagonist (Appendix A) were low compared to those in the liver.

### 2.7. No Difference in Nociceptive Behavior in Any of the Knockout Models

The pruriceptive behavioral studies suggested that peripheral LPA predominantly contributes to HCQ-evoked itch, suggesting that a specific subset of sensory neurons were LPA-sensitive. Previous studies revealed itch-specific populations of sensory neurons, but the majority of sensory neurons that respond to itch stimuli also contribute to nociceptive signaling. For example, TRPA1 positive neurons mediate itch and cold pain [65]. To address the specificity for itch in our models, we measured nociceptive responses upon stimulation with heat, cold and dynamic von Frey “hair” (Appendix A). There was no difference in any of the knockout mice versus the respective control group. Hotplate experiments of LPAR2^−/−^ were suggestive of a higher sensitivity towards nociceptive heat stimulation, but the apparent difference between genotypes was not confirmed in the Hargreaves tests (Appendix A, left panel). Hence, overall, we conclude that the candidate LPA receptors, LPA transporters and autotaxin are not involved in acute nociceptive responses.

### 2.8. Minor Reduction in Calcium Influx in DRG Neurons of Advillin-LPAR5^−/−^ Mice

To address the impact of LPAR5 on the itch sensation of primary sensory neurons, we assessed stimulated calcium influx in primary DRG neurons in primary cultures of Advillin-LPAR5^−/−^ mice versus floxed LPAR5-flfl mice in two sequential sets of experiments. In the first set, high K^+^ was used as final stimulus to elicit depolarization-evoked calcium currents. In the second set, capsaicin was used as final stimulus to address TRPV1-mediated calcium influx, which is a model for heat-evoked pain. About 10% of the primary DRG neurons were responsive to either histamine or HCQ, but not to both stimuli, whereby a positive response was defined as a minimum 1.3-fold increase in [Ca^2+^]_i_. The response to Cp 48/80 or HCQ was mutually exclusive. The fraction of neurons responding to either histamine or HCQ did not differ between genotypes (Figure 9). The fraction of responding neurons was also similar for high K^+^ and capsaicin. Looking at the time courses of the [Ca^2+^]_i_ curves, one can see a subtle reduction in HCQ-evoked peaks in the Advillin-LPAR5^−/−^ neurons (Figure 9), likely originating from the fraction of LPAR5 positive neurons. From these non-subpopulation-specific experiments, we cannot deduce a significant difference but nonetheless, these subtle results are in agreement with other behavioral studies and suggest that LPAs enhance stimulus-evoked itch via the LPAR5 stimulation of sensory neurons.

## 3. Discussion

In the present study, we show a segregated localization of LPAR5 in peripheral sensory neurons and LPAR2 in cortical and thalamic neurons and opposing effects of deletion of either LPAR5 or LPAR2 on HCQ-evoked itch responses. The deletion of LPAR5 in sensory neurons reduced itch, but the deletion of LPAR2 and of CNS-specific lipid-phosphate phosphatase, Prg1/LPPR4, enhanced HCQ-evoked itch. In contrast, nociception was not affected by any of the knockout models, suggesting a specific modulation of itch signaling. Figure 10 shows a hypothesis for putative LPA-dependent itch-signaling paths. LPAs have been previously shown to contribute to the sensation of itch, particularly, but not exclusively, to cholestatic itch [42,43,48]. Indeed, we observed a region-specific increase in LPAs in the skin and in the thalamus upon intradermal injection of a pruritogen. Peripheral LPAR5 was positioned to sense the release of LPA in the skin, while LPAR2 was positioned to respond to the central release. Based on previous electrophysiology studies in the hippocampus, we had expected excitatory effects of both receptors, LPAR5 and LPAR2. Indeed, LPAR5-deficient mice scratched less than the floxed controls. The aggravated response in the LPAR2^−/−^ mice suggests that the LPA-dependent sensory itch-evoked motor response circuit is complex. In the hippocampus, presynaptic LPAR2 and postsynaptic Prg1/LPPR4 acted as a homeostat to control LPA-dependent excitatory postsynaptic potentials (EPSCs). The deficiency of LPAR2 reduced EPSCs in case of Prg1 deficiency, whereas the deletion of Prg1 only increased EPSCs. It was concluded that synaptic LPAs drive glutamate release via presynaptic LPAR2 and are scavenged via postsynaptic Prg1 to maintain synaptic homeostasis [8]. The observed aggravated scratching responses in the Prg1 knockout mice agrees well with this circuit, but the enhanced scratching responses in the LPAR2 knockout mice rather suggested that LPAR2 activated heterosynaptic circuits [66], in analogy to the processing of lipid-mediated nociceptive processing in the spinal cord [67,68], or that LPAR2 is involved in the executive scratching pathways. Post in situ immunofluorescences studies with a marker of inhibitory neurons (GAD67) remained inconclusive, owing to the RNA-based techniques used to reveal synapses. LPAs may not only contribute to sensing the pruritogen via LPAR5 but also regulate itching–scratching cycles [69] and scratching-mediated itch relief and reward [70,71,72], possibly via LPAR2 (please see the drawing in the Graphical Abstract). The latter idea is supported by the high expression of LPAR2 in the paraventricular thalamus (PVT), which is involved in active defense mechanisms and the signaling of addictive pleasure [73,74,75].

To further address the peripheral versus central effects of LPAs, we compared the itching responses of mice with a GFAP-mediated deletion of astroglial autotaxin (central) versus those with pharmacologic autotaxin inhibition (systemic) [76]. Autotaxin produces LPAs from lysophosphatidylcholines (LPCs) at the extracellular sites of membranes [77]. The pharmacologic inhibition of autotaxin reduced itch in our studies, but the glial deletion of autotaxin had no effect. The result, again, suggested that the pro-itching effects of LPAs in healthy mice originate mainly in the periphery. Astrocytic autotaxin-mediated LPA generation likely requires a prior brain injury [61]. In agreement, LPAR5 inhibition reduced itch and partly reflected the behavior of Advillin-LPAR5^−/−^ mice. In contrast, LPAR2 blocking did not replicate the phenotype of the LPAR2^−/−^ mice. The drug may have failed pharmacokinetically to achieve concentrations high enough to inhibit synaptic LPAR2 within the time frame of the itching experiment. Alternatively, the pro-itch behavior of the LPAR2^−/−^ mice was due to adaptive mechanisms that do not occur upon short-lasting inhibition in single dose experiments. As the concentrations of the drug in the brain remained low, as compared to those in the plasma and liver (Appendix A), we favor the pharmacokinetic explanation.

Previous studies have shown that conventional LPAR5 knockout show reduced neuropathic pain-like behavior [39] and reduced scratching responses in itch models [42] similar to our study. The effects of the global LPAR5 knockout appears to be stronger [42] than the HCQ-restricted behavioral effects of our Advillin-LPAR5 knockout mice. Although inter-lab differences of behavioral assessments [78] may contribute to apparent differences in full versus Advillin-specific LPAR5 knockout mice, one may conclude that LPAR5 in keratinocytes contributes to the sensation of LPA-dependent itch, likely by releasing further pro-itching mediators. A contribution of keratinocytes would also explain the rather weak differences in calcium fluxes in the DRG cultures of Advillin-LPAR5^−/−^ versus LPAR5-flfl cultures, agreeing with a recent study, in which LPAR5-mediated signaling of keratinocytes was shown to drive the pathogenesis of psoriasis [79]. It is ofnote that only about ten percent of the DRG neurons responded to either histamine or HCQ, limiting the fraction of relevant neurons for our itch models. In hindsight, it might have been useful to employ a sequential stimulation of LPA18:1 or LPA20:4 and then histamine or HCQ to analyze only LPA-responsive neurons.

The observed strong expression of LPAR5 in the skin and in sensory neurons suggests that topical ointments containing LPAR5-antagonists might be useful therapeutically to reduce itch sensations that involve HCQ-dependent signaling paths. Cp 48/80-evoked responses were mostly not affected, suggesting that the histamine-evoked itch was not responsive or was less responsive to LPAs. The stimulus-dependent phenotypes of our knockout models may explain previous conflicting results showing that LPAs either did or did not contribute to non-cholestatic itch [43,47,48]. It is of note that HCQ-evoked itch-signaling shares receptors and signaling paths with neuropathic pain, such as the activation of TRPA1 [80,81], which also contributes to cold allodynia [42,48,82], whereas Cp 48/80 appears to be stronger related to inflammatory pain [83], with heat hypersensitivity predominated by TRPV channels [84,85]. The LPAR5 reporter mice revealed expression of LPAR5 in distinct P2X3-positive DRG neurons, but not in peptidergic CGRP- or SP-positive neurons. Its expression pattern did not match exactly with other markers and rather suggests that LPAR5-positive neurons constitute a mixed population.

In summary, we show that LPAs are released in the skin and thalamus upon intradermal injection of a pruritogen. LPAR5 mediates the sensation of the peripheral release while LPAR2 appears to be involved in the central mechanisms of scratching-mediated itch relief.

## 4. Materials and Methods

### 4.1. Mouse Lines

Heterozygous PRG1/LacZ reporter mice and PRG1^−/−^ mice were constructed and bred as described [8]. Prg1/LPPR4 gene knock out was achieved by replacing exons 4 to 5 with an IRES-β-Gal-Neo cassette. PRG2^−/−^ mice were generated by flanking exon-1 of PRG2/LPPR3 with loxP sites and subsequent deletion by crossing with pan-deletion Cre-mice [10]. LPAR2^−/−^ mice were obtained by replacing the second half of exon 2, including a partial sequence of the adjacent intron, with a neomycin cassette, as described [86]. GFAP-mediated deletion of autotaxin/ENPP2 was achieved by mating GFAP-Cre mice with floxed ENPP2-mice [56]. Sensory neuron specific LPAR5^−/−^ mice were generated by crossing LPAR5-flfl mice (EUCOMM) with Advillin Cre mice, similar to the procedure described in [87]. Briefly, floxed LPAR5 mice were obtained from the European Conditional Mouse Mutagenesis Program (EUCOMM; Lpar5^tm1a(KOMP)Wtsi^). These mice carry a conditional-ready allele, which was inserted by homologous recombination. In the construct, exon 2 of mouse LPAR5 (ENSMUSG00000067714) is flanked with loxP sites for Cre-mediated deletion. In front is a promoter-less, FRT-flanked LacZ-Neo reporter cassette, so that LacZ is expressed under the control of the LPAR5 promotor. LPAR5/LacZ mice were used to assess LPAR5 expression and localization. The reporter was then removed by breeding with Flp-mice and subsequently, LPAR5-flfl were crossed with Advillin-Cre mice to cut out exon 2 and create a sensory neuron-specific Advillin-LPAR5^−/−^ knockout mouse. The performance of Advillin-Cre mice has been described in detail elsewhere [88,89,90,91,92,93,94,95,96]. Genotyping followed the EUCOMM protocol (Primers in Appendix A). The successful deletion was confirmed at RNA level.

Experiments were performed with litter mates (LPAR3, PRG2) or age and sex-matched wildtype control mice, which were maintained on the respective background and raised in parallel (LPAR2, PRG1). PRG1, PRG2 and LPAR2 mice and respective controls were bred in the animal facility of the University of Mainz and were transported to Frankfurt at least one week before the start of the experiments. GFAP-ENPP2 mice and Advillin-LPAR5 mice were bred and maintained in Frankfurt. The sample sizes depended on the experiments and readouts. Groups comprised 8–22 mice per genotype at 15 to 30 weeks of age, as specified in the figure legends. Mice were allowed to acclimatize to the experiment rooms, cages or mazes before starting the experiments. They had free access to food and water and were maintained in climate-controlled rooms at a 12 h light–dark cycle.

The experiments were approved by the local Ethics Committee for Animal Research (Darmstadt, Germany V 54–19 c20/15–F 95/54, FK1080 and FK1110) and adhered to the European guidelines and to those of GV-SOLAS for animal welfare in science, as well as with the ARRIVE guidelines.

### 4.2. Quantitative Real-Time PCR

For tissue preparation, animals were sacrificed via carbon dioxide, followed by rapid blood withdrawal and decapitation. The brain and L4/5 spinal cord was dissected and rapidly frozen in liquid nitrogen. The brain was removed and sectioned coronally on a mouse-brain matrix to obtain 1 mm thick slices. The posterolateral and ventral thalamus and the somatosensory cortex were excised from the respective brain slice and rapidly frozen in liquid nitrogen and stored at −80 °C. The dissection time was monitored and did not exceed 5 min.

Total RNA was extracted from tissue according to standard procedures, using the Qiagen RNeasy Plus total RNA extraction kit, and was reverse-transcribed using the Verso DNA first-stand cDNA synthesis kit with poly-dT or random hexamers as primers. Quantitative rt-PCR was performed on an ABI 7500 Fast Real-time PCR System (Applied Biosystems, Darmstadt, Germany), using the SYBR green technique (Maxima SYBR Green/ROX qrt-PCR Master Mix; Thermo Fisher Scientific, Dreieich, Germany). Transcript regulation of LPAR2, LPAR5, MrgprA3, MrgprC11, Prg1, Prg2, ENPP2 and AGMO relative to the housekeeping gene, GAPDH, was determined using the relative delta Ct method, according to the manufacturer’s instructions (Applied Biosystems). Amplification was achieved at 60 °C for 35 cycles. Primer sequences are summarized in Appendix A.

### 4.3. Pruriception

Behavioral tests were performed by an investigator who was unaware of the mouse genotype or treatment and included 8 to 22 mice per group. Before starting, mice were acclimatized to the test chamber for at least 15 min. Itching behavior was measured using three pruritic stimuli: compound 48/80 (Cp 48/80, 50 µg in 100 µL saline, Sigma, Roedermark, Germany), hydroxychloroquine (HCQ, 200 µg in 100 µL saline, Sigma, Germany) and BAM 8–22 (Tocris, 100 µg in saline). Cp 48/80 evokes histamine-dependent itch, HCQ and BAM 8–22 primarily evoke histamine-independent itch responses. Cp 48/80, HCQ or BAM 8–22 was injected intradermally in the region of the nape of the neck, and the scratching response was observed for 30 min starting immediately after injection. Scratching times were summed in 5 min intervals. For tissue analyses of LPA concentrations mice were euthanized at the end of the itch observation period, the tissue was rapidly removed and snap-frozen on dry ice or liquid nitrogen and kept at −80 °C until analysis.

### 4.4. Drug Treatments

Drugs were injected intraperitoneally 20 min before stimulation of itching behavior. Doses were based on previous studies [1,56]. The following drugs were used: autotaxin inhibitor PF-8380 (Tocris; CAS 1144035-53-9), 10 mg/kg in 100 µL 5% DMSO/0.9% saline [76,97]; LPAR2 antagonist H2L5186303 (Tocris, CAS 139262-76-3), 30 µg/mouse (~1.2 mg/kg) in 100 µL 5% DMSO/0.9 % saline [1]; LPAR2 agonist GRI 977143 (Tocris, CAS 325850-81-5), 100 µg/mouse (~4 mg/kg) in 160 µL 5% DMSO/0.9% saline [1]; LPAR5 antagonist TC LPA5 4 (Tocris, CAS 1393814-38-4), 50 µg/mouse (~2 mg/kg) in 100 µL 5% DMSO/0.9% saline.

### 4.5. Nociception

Mice were habituated for 3 consecutive days to the test room, test cages and set-ups. The latency of paw withdrawal upon mechanical stimulation was assessed with a dynamic von Frey apparatus (Aesthesiometer, Ugo Basile, Gemonio, Italy), employing a force range of 0–5 g, a ramp of 0.5 g/s and holding at 5 g until paw withdrawal. The sensitivity to painful heat stimuli was assessed by recording the paw or tail withdrawal latency in the Hargreaves and tail flick tests (IITC Life Science, Woodland Hills, CA, USA), which employ a radiant heat source placed underneath the hind paw or the tail, with a mirror system. The lamp emits a heat beam until the paw is withdrawn or the tail is flicked. Sensitivity to heat and cold stimuli was also assessed by recording the paw withdrawal latency on a hot/cold plate (IITC Life Science). Hot-plate testing included measurements at a constant temperature (52 °C,) with a cut-off time of 40 s and a dynamic temperature increase (25–55 °C; 0.1 °C/s; cut-off 60 s). Dynamic cold-plate testing was conducted similarly (30–0 °C; 0.1 °C/s; hold at 0 °C, cut-off of 60 s). Tests were performed two or three times per mouse, with an inter-trial interval of at least 20 min.

### 4.6. Primary Neuron Culture

Primary neuron-enriched cultures of dorsal root ganglia (DRG) neurons were prepared by dissecting DRGs of adult mice into Hank’s balanced salt solution (Dulbecco) (Invitrogen, Waltham, MA, USA) with 10 mM 4-(2-hydroxyethyl)-1-piperazine ethane sulfonic acid (HEPES) (Invitrogen), followed by digestion with 2.5 mg/mL collagenase A (Millipore) and 1 mg/mL dispase II (Invitrogen) before short treatment with DNase (Sigma, Germany, 250 U per sample). Triturated cells were centrifuged through a 15% fat-free bovine serum albumin solution, plated and cultivated on poly-L-lysine-coated cover slips in a serum-free Neurobasal medium (Gibco-BRL) containing 1x B27 supplement (Gibco-BRL, Waltham, MA, USA), 1% penicillin/streptomycin (Sigma-Aldrich), 200 ng/mL nerve growth factor (Gibco-BRL) and 2 mM L-glutamine (Gibco-BRL), at 37 °C and 5% CO_2_. Neurons were subjected to calcium imaging 24 h after plating.

### 4.7. Calcium Imaging

Calcium influx was measured in primary sensory neuron cultures upon stimulation with histamine, hydroxychloroquine, capsaicin and high K^+^. Capsaicin was used as a positive control. It evokes calcium influx via activation of the transient receptor potential channels, TRPV1. High K^+^ was used to assess depolarization-evoked calcium currents. Calcium influx in response to high K^+^ is a criterion of neuronal viability. Calcium-imaging experiments were performed with a Leica calcium-imaging setup. Images were captured every two seconds. Cells were loaded with 5 μM of the Ca^2+^-sensitive fluorescent dye Fura-2-AM-ester, incubated for 40 min at 37 °C and washed three times with Ringer’s solution. Coverslips were then transferred to a perfusion chamber with a flow rate of 1–2 mL/min, at room temperature, and calcium fluxes were measured fluorometrically as the ratio of the absorbances at 340 and 380 nm (F 340/380). Baseline ratios were recorded for 100–180 s, before application of 1 mM histamine (180–240 s), followed by a subsequent wash out with Ringer’s solution (240–600 s), then application of 0.5 mM hydroxychloroquine from 600–630 s, wash out again (630–1020 s), final 100 nM capsaicin (1020–1050 s) to activate TRPV1 ion channels, or final 75 mM KCl (1020–1080 s) to evoke depolarization-mediated calcium currents. The final stimulus was again washed out with Ringer and cells were observed for up to 1200 s. Data are presented as changes in fluorescence ratios (F340/380) normalized to baseline ratios. The analysis encompassed 300–500 neurons per genotype and experiment, with 6–9 independent DRG cultures, each from eight mice per group per stimulus. Mice were 15–20 weeks old at sacrifice. Maximum fold increase, time to maximum and area of the fold increase versus time curve were calculated by using the linear trapezoidal rule. The time courses and areas were used for statistical comparison.

### 4.8. Histology and Tissue Immunofluorescence

Mice were terminally anesthetized with isoflurane and transcardially perfused with cold 0.9% NaCl, followed by 4.5% phosphate-buffered paraformaldehyde (PFA) for fixation. Tissues were excised, post-fixed in 4.5% buffered PFA (pH 7.4) for 2 h, cryoprotected overnight in 20% sucrose at 4 °C, embedded in small tissue molds in cryo-medium and cut on a cryotome at 10 μm for DRGs or 12 µm for spinal cord and brain sections.

For immunofluorescence, slides were permeabilized in 1 × PBS with 0.1% Triton-X-100 (PBST), then blocked with 3% BSA or 5% normal goat serum (NGS) in PBST for 30–60 min at room temperature and subsequently incubated with primary antibodies in 1% BSA or 1% NGS in PBST, overnight, at 4 °C. Primary antibodies are listed in Appendix A. After washing, slides were incubated with the secondary antibody labeled with a fluorochrome for 2 h, at room temperature, followed by washing in PBS and 10 min incubation with DAPI in PBS at room temperature. The settings were optimized for the respective antibodies and tissues.

For X-Gal/FDG staining, Prg1/LacZ or LPAR5/LacZ reporter mice were used. Tissue sections were washed in 1 × PBS, shortly rinsed in distilled water and subjected to detection of beta-galactosidase using staining with X-Gal (Sigma-Aldrich; 4 mg/mL) or with Fluorescein di-β-D-galactopyranoside, FDG (F1179, Life technologies; 200 µM). X-Gal or FDG was diluted in staining buffer consisting of 5 mM potassium ferrocyanide, 5 mM potassium ferricyanide, 0.01% (*w*/*v*) sodium deoxycholate, 1 mM MgCl2 and 0.02% (*v*/*v*) Nonidet P40. Slides were incubated with the FDG staining solutions for 10 min (brain) or for 3–5 h (spinal cord). X-Gal staining was performed over night at 37 °C. Slides were then washed in cold PBS, followed by short rinsing in cold distilled water. FDG staining was followed by immunostaining, as described above. Finally, slides were embedded in Aqua-Poly/Mount.

For skin histology, a shaved area of the neck skin was excised. Skin sections were dehydrated and embedded in paraffin and cut on a microtome (3.5 µm). Paraffin-embedded sections were deparaffinized in xylene and graded ethanol and stained with hematoxylin and eosin (HE) or with specific antibodies, followed by DAB staining.

Images were taken with an inverted fluorescence microscope (Axiovert 200, Carl Zeiss, Jena, Germany) or Zeiss LSM510 Meta inverted confocal laser-scanning microscope. Overview images were taken with a Keyence BZ-9000 fluorescence microscope in bright-field or fluorescence mode.

### 4.9. In Situ Hybridization in Tissue Slices with RNA Scope

In situ hybridization was performed by using a colorimetric RNA scope 2.5 HD Detection kit (Advanced Cell Diagnostics) with custom-made probes targeting LPAR2 (VB1-12913-VC) mRNA. Terminally anesthetized mice were perfused transcardially with 0.9% saline and 4% phosphate-buffered PFA. Then, the brain, DRGs and spinal cord were excised and post-fixated for 3 h at 21–23 °C, using a 4% PFA solution with 5.4% glucose (wt/vol), 0.01 M sodium metaperiodate in lysine-phosphate buffer. For cryoprotection, tissues were incubated in 20% (*w*/*v*) sucrose in DEPC-treated 1 × PBS overnight at 4 °C. The tissue was cryo-sectioned at 12 μm thickness and the sections were subjected to in situ hybridization according to the manufacturer’s protocol, skipping the dehydration/rehydration and proteinase QS treatment steps. Sections were incubated at 40 °C (3–5 h), washed, then hybridized with pre-amplification oligos (1:100) at 40 °C for 25 min, washed, hybridized with amplification oligos (1:100) at 40 °C for 15 min, washed, and finally hybridized with the label oligos (1:100) carrying horseradish peroxidase (HRP) at 40 °C for 15 min. Slides were developed with the HRP substrate, RED RNA scope 2.5 HD Detection Reagent (Advanced Cell Diagnostics, Newark, CA, USA) and counterstained with hematoxylin. After completion of the in situ hybridization, immunofluorescent staining was added in some experiments, following the procedures described above. Blocking, staining and washing buffers contained an RNase inhibitor (Promega, Madison, WI, USA, RNasin Plus RNase inhibitor, 1:1000). Images were taken with a Keyence BZ-9000 fluorescence microscope in the bright-field mode.

### 4.10. Extraction and Analysis of Lysophosphatidic Acids (LPAs)

Lysophosphatidic acids were extracted from mouse plasma, brain and skin tissue with chloroform, after adding the internal standard LPA 17:0 (IS). The extraction mixture consisted of 50 µL plasma or homogenized tissue, 10 µL methanol, 20 µL IS, 800 µL 0.1 M HCl/MeOH 1:1 (*v*/*v*) and 400 µL chloroform. The organic layer was removed, the sample was evaporated at 45 °C under a gentle stream of nitrogen, and reconstituted in 200 µL methanol. For calibration standards and QC samples, 50 µL PBS were spiked with 10 µL standard solution and processed as described above. Quantitative analysis was performed with liquid chromatography–electrospray ionization–tandem mass spectrometry (LC-ESI-MS/MS), as described with minor modifications [1,98]. The LC-MS/MS system consisted of a quadrupole-ion trap mass spectrometer QTrap 5500 (Sciex, Darmstadt, Germany), equipped with a Turbo-V-source operating in negative ESI mode and an Agilent 1260 HPLC-system.

For the chromatographic separation, a C18 column (Mercury 20 × 2.0 mm, 3 µm) and precolumn were used (both from Phenomenex, Aschaffenburg, Germany). A linear gradient was employed at 400 µL/min. Mobile phase A was water with 50 mM ammonium formate and formic acid (100:0.2, *v*/*v*), while mobile phase B was acetonitrile/isopropanol with formic acid (50:50:0.2, *v*/*v*/*v*). The total run time was 7 min, and the injection volume 15 µL. The mass spectrometer operated in the negative ion mode with an electrospray voltage of −4500 V at 350 °C. Multiple reaction monitoring (MRM) was used for quantification. Data acquisition was performed using Analyst Software V1.6.2 (Sciex) and the quantification was performed using MultiQuant V3.0.2 (Sciex), using the internal standard method. Quality controls (low, medium, high) were included at the beginning and end of each run. Calibration curves were calculated by linear regression with 1/x weighting. Variations in accuracy of the calibration standards were less than 15% over the calibration range, except for the lower limit of quantification, where a variation in accuracy of 20% was accepted. For the acceptance of the analytical run, the accuracy of the QC samples had to be between 85% and 115% of the nominal concentration for at least 67% of all QC samples.

### 4.11. Statistics

The numbers of animals used for the experiments are indicated in the respective figure legends. Quantitative data are presented as box/scatter plots, where each scatter represents one mouse. Histologic experiments were performed with three mice for RNAscope, and with 11–18 reporter mice (Prg1-LacZ, LPAR5-LacZ). Group data are presented as mean ± SD or mean ± sem, with the latter used for behavioral time course data, specified in the respective figure legends. Data were analyzed with SPSS 27 and Graphpad Prism 8 or 9 and Origin Pro 2022. Data were mostly normally distributed, or log-normally distributed, according to the Shapiro–Wilk test. For testing the null-hypothesis that the groups were identical, the means of two groups were compared with 2-sided, unpaired Student’s *t*-tests. Time course data or multifactorial data were submitted to 2-way analysis of variance (ANOVA) using, e.g., the factors ‘time’ and ‘genotype’. In case of significant differences, groups were mutually compared at individual time points using post hoc *t*-tests according to Dunnett or according to Šidák. In case of violations of sphericity, degrees of freedom were adjusted according to Huynh Feldt. Asterisks in figures show multiplicity-adjusted p-values. The areas under the curve of calcium imaging data were Log2-transformed. The areas were calculated by using the linear trapezoidal rule.

## Figures and Tables

**Figure 1 ijms-25-08177-f001:**
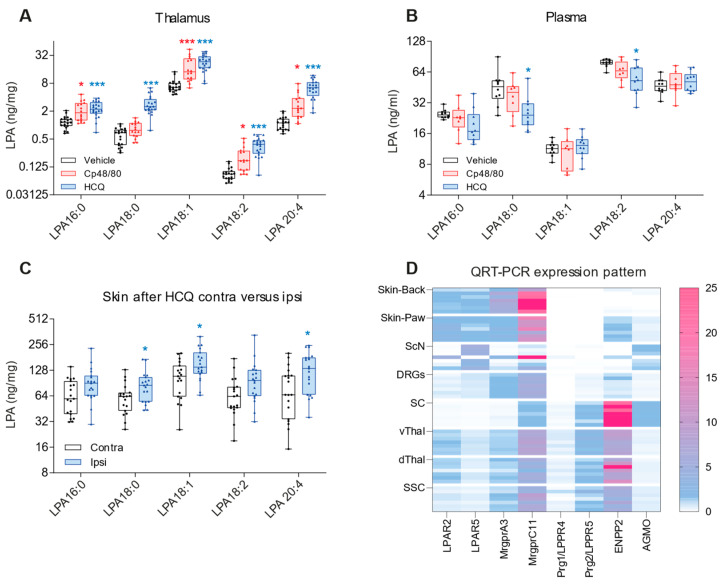
LPA release in the skin and thalamus evoked by itch stimulation. (**A**) Concentrations of LPA species in the thalamic region after intradermal injection of saline (vehicle), Cp 48/80 or HCQ. Tissue was collected 30–45 min after injection. The box is the interquartile range, whiskers show minimum and maximum, and scatters represent the mice. Each mouse is depicted as two scatters of each two samples. (**B**) Plasma LPAs of the mice shown in A, n = 8–10 per group. (**C**) Concentrations of LPA species in the skin on the side of the HCQ injection (ipsilateral) and the opposite side (contralateral). The scatters represent individual mice. Data were compared with a two-way ANOVA and subsequent post hoc *t*-test using an adjustment of alpha according to Šidák. * *p* < 0.05, *** *p* < 0.001. (**D**) Heatmap of mRNA expression of candidate genes involved in itch signaling in the CNS and periphery. The color lines represent the mice (n = 7). Abbreviations: ScN, sciatic nerve; DRG, dorsal root ganglia; SC, spinal cord; vThal, ventral thalamus; dThal, dorsal thalamus; SSC, somatosensory cortex. MRGPR, mas-related G protein-coupled receptor; AGMO, alkylglycerolmonooxygenase.

**Figure 9 ijms-25-08177-f009:**
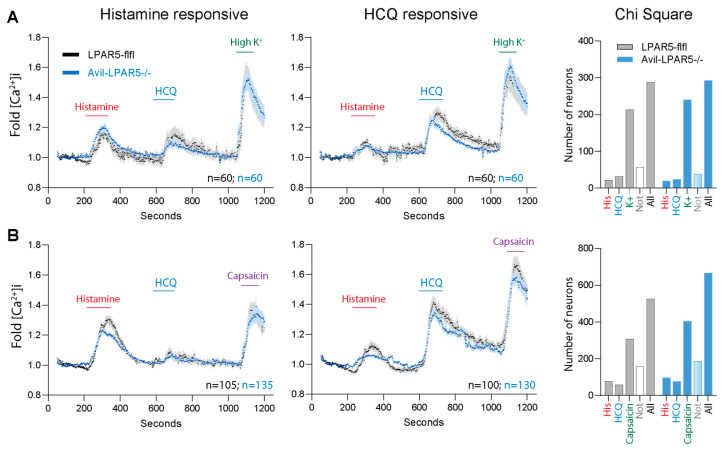
Histamine and HCQ-evoked calcium fluxes in primary sensory neurons of Advillin-LPAR5^−/−^ mice. Calcium imaging was performed by analysis of absorbance ratios of Fura-2 upon excitation at 340 and 380 nm. Primary DRG neurons of Advillin-LPAR5^−/−^ and LPAR5-flfl control mice were stimulated sequentially with 1 mM histamine, 0.5 mM HCQ and final 75 mM K^+^ (in **A**) or final 0.1 µM capsaicin (in **B**). The left panels show the time courses of the fold change in [Ca^2+^]_i_ versus baseline. The Chi-square contingency analysis shows the total numbers of neurons and stimulus-responsive neurons. A response was defined as a fold increase in [Ca^2+^]_i_ > 1.3. Time courses show responsive neurons. “Not” means not responsive. (**A**) Calcium imaging using a final stimulus of high potassium (75 mM K^+^) to evoke depolarization-dependent calcium influx. (**B**) Calcium imaging using capsaicin as final stimulus to evoke TRPV1-dependent calcium influx.

**Figure 10 ijms-25-08177-f010:**
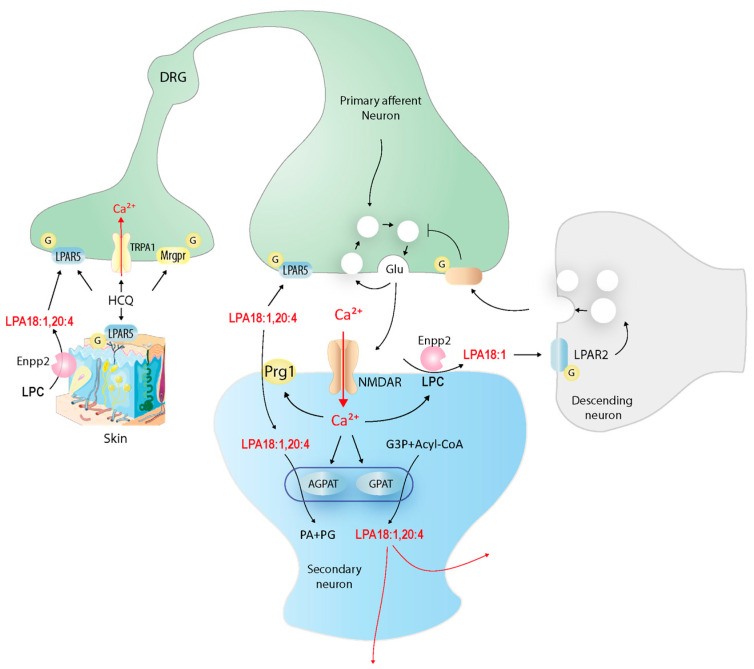
Putative LPA-pruriceptive signaling paths. LPAs are released from keratinocytes upon pruriceptive stimulation and activate fiber terminals of LPAR5-positive pruriceptive neurons, causing glutamate or neuropeptide release from central terminals and activation of postsynaptic neurons. Calcium influx stimulates LPA generation and release from secondary neurons. Extracellularly, LPA is generated from LPC via autotaxin (ENPP2). Synaptic LPA can be recycled via Prg1/LPPR4.

## Data Availability

Data generated in this study are presented in the main body of this manuscript or the Appendix A. Additional raw data are available from the corresponding author on reasonable scientific request.

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
