# Peer review of "Lysophosphatidic Acid Receptors LPAR5 and LPAR2 Inversely Control Hydroxychloroquine-Evoked Itch and Scratching in Mice"

_ijms, 2024, doi:10.3390/ijms25158177_

Round 1
Reviewer 1 Report
Comments and Suggestions for Authors
The paper deals with experimental evidence for the role of LPA and their receptors LPAR 5 and LPAR2 in non-histaminergic-induced scratching behavior. Although the authors do not present complete evidence for the hypothesis - the study is extensive and worth publishing. The following corrections and additions are suggested:
1. in order to make your paper easier to follow, please in the introduction draw a scheme of synthesis, sites of action and degradation of LPA, which contains standard and alternative names of enzymes and receptors involved in signaling pathways (LPAs, LPAR2 and LPAR5, phospholipases, lysophospholipid phosphatases). You can add agonists and antagonists of enzymes and receptors to the same scheme. Also, better explain the role of Prg1/LPPR4 and Prg2/LPPR3.
2. When you say in introduction that the selection of LPAR5, LPAR2, Prg1, Prg2 and autaxin for this study is based on previous work and mRNA studies, do you mean your work or the work of other scientists - please provide references.
3. When you say in the results that itch stimulation led to an increase in LPA in the thalamus and skin, but not in the serum, please also state how you caused itch stimulation. In the materials and methods there is a description of the extraction and analysis of LPAs, but there is no description of the stimulation and tissue collection. Also, in Figure 1, you do not state on how many samples you based your analysis on, so it cannot be concluded whether the sample was sufficient for statistical analysis. Can you explain the statistically significant drop in LPA 18:0 and LPA 18:2 after stimulation with HCQ?
4. In all Figures, indicate the exact number of samples used for individual analyses.
5. In Figure 2A showing the X-Gal histology results, for those unfamiliar with skin histology, explain which structures the blue signals correspond to. You say that you did not detect the LPAR5 signal on the terminal nerve fibers, but you did not show where the terminal nerve fibers are located in the image. Can you explain why it is enough that LPAR5 is on DRG bodies and not on nerve terminals to be associated with itch. How production of LPA is stimulated in DRG?
6. Can you quantify the mRNA expression results in Figure 4? Explain why you didn't do the same analysis for skin or DRG?
7. Is there a reason why you did not show stimulation with BAM 8-22 for the LPAR5 knockout?
8. The title 2.5 (line 221) does not fully correspond to the results shown in Figure 7 because you marked statistical significance on the graph with the results of HCQ stimulation for both mouse models.
9. Are you sure that the autotaxin inhibitor does not have off-targeted effects?
10. Figure 10 is missing.
11. In the abstract, emphasize that this study has potential translational significance in the context of psoriasis.
12. In materials and methods, write how many animals were used in which part of the study. 13. Where did you use primary neuron cultures explained in Materials and methods?
Author Response
Reviewer #1
The paper deals with experimental evidence for the role of LPA and their receptors LPAR 5 and LPAR2 in non-histaminergic-induced scratching behavior. Although the authors do not present complete evidence for the hypothesis - the study is extensive and worth publishing. The following corrections and additions are suggested:
We thank you for your time for evaluation of our manuscript and helpful comments. The responses to your suggestions are detailed below.
- in order to make your paper easier to follow, please in the introduction draw a scheme of synthesis, sites of action and degradation of LPA, which contains standard and alternative names of enzymes and receptors involved in signaling pathways (LPAs, LPAR2 and LPAR5, phospholipases, lysophospholipid phosphatases). You can add agonists and antagonists of enzymes and receptors to the same scheme. Also, better explain the role of Prg1/LPPR4 and Prg2/LPPR3.
We have added a scheme showing the hypothetical signaling path of LPA at the sensory synapse as requested. The figure is now Figure 10. As it is based on the results of the present manuscript it is placed at the end of the Discussion. For a more systemic view we would also like to refer to the Graphical Abstract.
The metabolism and receptors of LPAs have been extensively reviewed, and multiple images have been posted and are freely available in e.g. google images. We feel that an extensive introduction of the localization of LPA receptors and metabolizing enzymes is beyond the scope of the present Original Research paper. It is rather a topic of a review. We have added references of recent reviews ([1-4]).
- When you say in introduction that the selection of LPAR5, LPAR2, Prg1, Prg2 and autaxin for this study is based on previous work and mRNA studies, do you mean your work or the work of other scientists - please provide references.
References are now added to the sentence. mRNA studies are shown in Figure 1D.
- When you say in the results that itch stimulation led to an increase in LPA in the thalamus and skin, but not in the serum, please also state how you caused itch stimulation. In the materials and methods there is a description of the extraction and analysis of LPAs, but there is no description of the stimulation and tissue collection. Also, in Figure 1, you do not state on how many samples you based your analysis on, so it cannot be concluded whether the sample was sufficient for statistical analysis. Can you explain the statistically significant drop in LPA 18:0 and LPA 18:2 after stimulation with HCQ?
The itch stimulation is described in Methods under 4.3 Pruriception. A sentence describing tissue collection is now added. Figure 1 shows LPAs in thalamus and plasma in mice stimulated with vehicle, Cp48/80 (histamine evoked) and HCQ (chloroquine evoked). Skin samples were only analyzed after HCQ versus vehicle. The figures show (different colored boxes) what the stimulus was. All quantitative results in this manuscript are presented as box/scatter plots where each scatter is one mouse (n = 8-10). The sample sizes are detailed in the figure legends. It is rechecked and all information was already given. The heatmap in figure 1 (rtPCR) is based on n = 7 animals per group as explained in the figure legend. Each color line in one mouse. We have highlighted the information in yellow. It might have been overlooked.
Drop of LPA18:0 and LPA18:2 in plasma after HCQ: It might be stress induced. But we do not know.
- In all Figures, indicate the exact number of samples used for individual analyses.
All quantitative results in this manuscript are presented as box/scatter plots where each scatter is one mouse. The figure legends were rechecked and the requested information was already given.
For the qualitative histology evaluation of the localization we used 11 Prg1-LacZ mice, 18 LPAR5-LacZ mice, 3 mice for RNAscope, and 12 mice for antibody based histology of LPAR2 in LPAR2-/- (n = 6) versus LPAR2+/+ mice (n = 6). Antibody-based LPAR2 studies were unspecific and are not shown. The information is now added to the figure legends and in Methods 4.11 Statistics.
- In Figure 2A showing the X-Gal histology results, for those unfamiliar with skin histology, explain which structures the blue signals correspond to. You say that you did not detect the LPAR5 signal on the terminal nerve fibers, but you did not show where the terminal nerve fibers are located in the image. Can you explain why it is enough that LPAR5 is on DRG bodies and not on nerve terminals to be associated with itch. How production of LPA is stimulated in DRG?
The figure legend in the Figure shows Eosin in pink and X-Gal in blue to reveal what the blue stain is. The stain is explained in the figure. In addition, it is now added in the legend.
In the reporter mice, the cDNA for beta galactosidase in under control of the LPAR5 promoter so that the reporter, beta galactosidase, is only produced in cells where the LPAR5 promoter is active. However, transport of beta galactosidase protein along axons and insertion into membranes is not identical to LPAR5. That we did not see X-Gal stain (i.e. activity of beta galactosidase reporter) in the nerve terminals does not mean that LPAR5 is not at the nerve terminal and contained in soma only. Indeed, mRNA analyses in Figure 1D show high LPAR5 in the sciatic nerve.
- Can you quantify the mRNA expression results in Figure 4? Explain why you didn't do the same analysis for skin or DRG?
The RNAscope experiments were meant for localization of LPAR2 not to quantify the numbers of positive neurons or relative intensities at different sites.
DRGs: Figure 1D shows at mRNA (rtPCR) level that there is no/almost no LPAR2 mRNA in the DRGs. Therefore RNAscope was not done with DRG sections.
Skin: We have not established RNAscope in the skin.
- Is there a reason why you did not show stimulation with BAM 8-22 for the LPAR5 knockout?
We had not enough mice for the experiment, and then Corona interfered with colony maintenance and the PhD student left.
- The title 2.5 (line 221) does not fully correspond to the results shown in Figure 7 because you marked statistical significance on the graph with the results of HCQ stimulation for both mouse models.
The title 2.5 is now changed to "mostly no effect" of PRG2 knockout
- Are you sure that the autotaxin inhibitor does not have off-targeted effects?
Off-target effects can never be excluded. So far, off target effects have not been revealed for PF8380.
- Figure 10 is missing.
Figure 10 is now added and is the scheme requested in comment-1.
- In the abstract, emphasize that this study has potential translational significance in the context of psoriasis.
It might be relevant for psoriasis but we have not studied psoriasis in a psoriasis model. Hence, we prefer not to speculate in the Abstract in terms of translational significance that goes beyond our results.
- In materials and methods, write how many animals were used in which part of the study.
In Methods 4.11 Statistics it is added that all quantitative results in this manuscript are presented as box/scatter plots where each scatter is one mouse. The figure legends detail the numbers of mice in the respective figure. Qualitative histology was done with 11-18 reporter mice and three wildtype for RNAscope.
- Where did you use primary neuron cultures explained in Materials and methods?
For calcium imaging (Figure 9).
References
- Zhou, Y.; Little, P. J.; Ta, H. T.; Xu, S.; Kamato, D., Lysophosphatidic acid and its receptors: pharmacology and therapeutic potential in atherosclerosis and vascular disease. Pharmacol Ther 2019, 204, 107404.
- Yanagida, K.; Shimizu, T., Lysophosphatidic acid, a simple phospholipid with myriad functions. Pharmacol Ther 2023, 246, 108421.
- Yaginuma, S.; Omi, J.; Kano, K.; Aoki, J., Lysophospholipids and their producing enzymes: Their pathological roles and potential as pathological biomarkers. Pharmacol Ther 2023, 246, 108415.
- Dacheux, M. A.; Norman, D. D.; Tigyi, G. J.; Lee, S. C., Emerging roles of lysophosphatidic acid receptor subtype 5 (LPAR5) in inflammatory diseases and cancer. Pharmacol Ther 2023, 245, 108414.

Reviewer 2 Report
Comments and Suggestions for Authors
It is a very interesting work performed by Fischer and colleagues, but there are some little issues that must be changed before acceptance.
It does not make sense finishing the first paragraph of results with this sentence “and the localization of LPA receptors (LPAR2, LPAR5; Fig 2-4) and transporters (Prg-1, Figure 5) at crucial synaptic sites.”, since you are going to discuss this data later on the text when you in fact present the data.
How did you reached this conclusion “The release pattern suggested that LPAs in the skin would lead to activation of itch-sensitive DRG neurons, and LPAs in the thalamus affect the thalamic control of cortical stimulation”?
There is no commercial antibody against LPAR5 to be tested on the skin to observe receptor presence? It will be interesting performing a double immunofluorescence for LPAR5 and PGP9.5 to see if there is a colocalization with the primary sensory fibers.
Since the presence of LPAR2 was done by RNAscope, you should do the same for LPAR5. This is an elegant technique that will increase the results quality.
A control for the Advillin-LPAR5-/- mice deletion must be presented. With RNAscope or immunofluorescence.
For figure 6 and 7 there is no indication of which one is letter a, b…
Author Response
Reviewer #2
It is a very interesting work performed by Fischer and colleagues, but there are some little issues that must be changed before acceptance.
We thank you for your time for evaluation of our manuscript and your helpful comments. The responses to your suggestions and questions are detailed below.
It does not make sense finishing the first paragraph of results with this sentence “and the localization of LPA receptors (LPAR2, LPAR5; Fig 2-4) and transporters (Prg-1, Figure 5) at crucial synaptic sites.”, since you are going to discuss this data later on the text when you in fact present the data.
Line 100-102 briefly outlines where the respective results are to be found. We think it helps navigation through the paper.
How did you reached this conclusion “The release pattern suggested that LPAs in the skin would lead to activation of itch-sensitive DRG neurons, and LPAs in the thalamus affect the thalamic control of cortical stimulation”?
The suggestion (not conclusion) is based on the analysis of LPA release at the respective sites and the localization of the receptors. It is not "proof" but "suggestion" as stated in the sentence.
There is no commercial antibody against LPAR5 to be tested on the skin to observe receptor presence? It will be interesting performing a double immunofluorescence for LPAR5 and PGP9.5 to see if there is a colocalization with the primary sensory fibers.
There is indeed no LPAR5 antibody that works for histology/immunofluorescence. In particular, staining of sensory nerve terminals in the skin requires that it works with paraffin embedded tissue. We have tried, but is was not successful.
Since the presence of LPAR2 was done by RNAscope, you should do the same for LPAR5. This is an elegant technique that will increase the results quality.
We have no RNAscope results for LPAR5. LPAR5 localization was done with LPAR5-LacZ reporter mice which is more reliable than RNAscope.
A control for the Advillin-LPAR5-/- mice deletion must be presented. With RNAscope or immunofluorescence.
We have no RNAscope results for LPAR5 as stated above and antibodies against LPAR5 do not work for immunofluorescence at least not in our hands. We and others have described the performance of Advillin-Cre mice in detail previously [1-9]. The references are now added. For this study, we only confirmed the expected deletion in sensory neurons at rtPCR level.
For figure 6 and 7 there is no indication of which one is letter a, b…
Column headlines in Figures 6 and 7 show the mouse lines and row headlines show the stimuli. This is now explicitly told in the legend. Adding A, B…. does not increase the clarity of the figures. Therefore, we have not added letters to subpanels.
References
- Valek, L.; Tran, B. N.; Tegeder, I., Cold avoidance and heat pain hypersensitivity in neuronal nucleoredoxin knockout mice. Free Radic. Biol. Med. 2022, 192, 84-97.
- Vogel, A.; Ueberbach, T.; Wilken-Schmitz, A.; Hahnefeld, L.; Franck, L.; Weyer, M. P.; Jungenitz, T.; Schmid, T.; Buchmann, G.; Freudenberg, F.; Brandes, R. P.; Gurke, R.; Schwarzacher, S. W.; Geisslinger, G.; Mittmann, T.; Tegeder, I., Repetitive and compulsive behavior after Early-Life-Pain associated with reduced long-chain sphingolipid species. Cell Biosci. 2023, 13, (1), 155.
- Latremoliere, A.; Latini, A.; Andrews, N.; Cronin, S. J.; Fujita, M.; Gorska, K.; Hovius, R.; Romero, C.; Chuaiphichai, S.; Painter, M.; Miracca, G.; Babaniyi, O.; Remor, A. P.; Duong, K.; Riva, P.; Barrett, L. B.; Ferreiros, N.; Naylor, A.; Penninger, J. M.; Tegeder, I.; Zhong, J.; Blagg, J.; Channon, K. M.; Johnsson, K.; Costigan, M.; Woolf, C. J., Reduction of Neuropathic and Inflammatory Pain through Inhibition of the Tetrahydrobiopterin Pathway. Neuron 2015, 86, (6), 1393-406. doi: 10.1016/j.neuron.2015.05.033.
- Chuang, Y. C.; Lee, C. H.; Sun, W. H.; Chen, C. C., Involvement of advillin in somatosensory neuron subtype-specific axon regeneration and neuropathic pain. Proc. Natl. Acad. Sci. U. S. A. 2018, 115, (36), E8557-e8566.
- Minett, M. S.; Nassar, M. A.; Clark, A. K.; Passmore, G.; Dickenson, A. H.; Wang, F.; Malcangio, M.; Wood, J. N., Distinct Nav1.7-dependent pain sensations require different sets of sensory and sympathetic neurons. Nat Commun 2012, 3, 791.
- Zurborg, S.; Piszczek, A.; Martínez, C.; Hublitz, P.; Al Banchaabouchi, M.; Moreira, P.; Perlas, E.; Heppenstall, P. A., Generation and characterization of an Advillin-Cre driver mouse line. Mol Pain. 2011, 7:66., (doi), 10.1186/1744-8069-7-66.
- Lau, J.; Minett, M. S.; Zhao, J.; Dennehy, U.; Wang, F.; Wood, J. N.; Bogdanov, Y. D., Temporal control of gene deletion in sensory ganglia using a tamoxifen-inducible Advillin-Cre-ERT2 recombinase mouse. Mol. Pain 2011, 7, 100.
- Pagadala, P.; Park, C. K.; Bang, S.; Xu, Z. Z.; Xie, R. G.; Liu, T.; Han, B. X.; Tracey, W. D., Jr.; Wang, F.; Ji, R. R., Loss of NR1 subunit of NMDARs in primary sensory neurons leads to hyperexcitability and pain hypersensitivity: involvement of Ca(2+)-activated small conductance potassium channels. J. Neurosci. 2013, 33, (33), 13425-30.
- Hasegawa, H.; Abbott, S.; Han, B. X.; Qi, Y.; Wang, F., Analyzing somatosensory axon projections with the sensory neuron-specific Advillin gene. J. Neurosci. 2007, 27, (52), 14404-14.

Round 2
Reviewer 2 Report
Comments and Suggestions for Authors
The authors replied to all my questions and I have no further suggestion.